# Complex Roles of Neutrophils during Arboviral Infections

**DOI:** 10.3390/cells10061324

**Published:** 2021-05-26

**Authors:** Abenaya Muralidharan, St Patrick Reid

**Affiliations:** Department of Pathology and Microbiology, University of Nebraska Medical Center, Omaha, NE 68198-5900, USA; abenaya.muralidharan@unmc.edu

**Keywords:** neutrophils, arboviruses, mosquito, inflammation, pathology

## Abstract

Arboviruses are known to cause large-scale epidemics in many parts of the world. These arthropod-borne viruses are a large group consisting of viruses from a wide range of families. The ability of their vector to enhance viral pathogenesis and transmission makes the development of treatments against these viruses challenging. Neutrophils are generally the first leukocytes to be recruited to a site of infection, playing a major role in regulating inflammation and, as a result, viral replication and dissemination. However, the underlying mechanisms through which neutrophils control the progression of inflammation and disease remain to be fully understood. In this review, we highlight the major findings from recent years regarding the role of neutrophils during arboviral infections. We discuss the complex nature of neutrophils in mediating not only protection, but also augmenting disease pathology. Better understanding of neutrophil pathways involved in effective protection against arboviral infections can help identify potential targets for therapeutics.

## 1. Introduction

Neutrophils are the most abundant leukocytes in the blood. They serve as the first line of defense against incoming pathogens, quickly mobilizing to the site of infection [1]. While neutrophils can have protective immunostimulatory activities, they can also have debilitating immunosuppressive activities by inhibiting T cell functions [1,2,3]. In addition, some viruses such as influenza, specifically H5N1, and West Nile virus are known to infect and replicate within neutrophils, using these cells as reservoirs for dissemination, although mechanisms involved in this phenomenon remain unclear [4,5].

Arthropod-borne viruses or arboviruses have a unique effect on neutrophil function since viral factors as well as vector factors can affect the activity of these cells. Arboviruses are a diverse group of viruses [6,7,8]. They are transmitted through blood-feeding insects such as mosquitoes and ticks, and include viruses such as chikungunya virus (CHIKV), dengue virus (DENV), West Nile virus (WNV), Zika virus (ZIKV), yellow fever virus (YFV), and eastern equine encephalitis virus (EEEV). This group of viruses have been known to cause significant morbidity and mortality around the world, with the potential to spread quickly and expand their geographical range due to their distinct mode of transmission using arthropods. Furthermore, the changes in climate and increase in urbanization help augment transmission and infectivity of these viruses [9,10,11,12,13,14,15]. 

The heterogeneity among this exceptionally large group of viruses makes developing therapeutics challenging. Neutrophils are generally the first to infiltrate infected sites. However, the role of these cells during viral infection is not yet fully understood [16]. Targeting neutrophil pathways or proteins that activate or suppress neutrophils can serve as a useful strategy for drug and vaccine development. In this review, we highlight the recent advancements in understanding the beneficial and detrimental nature of neutrophils during arboviral infections.

## 2. Mosquitoes

Ticks and mosquitoes enhance disease severity as well as transmission of many viruses [17,18,19,20,21]. Mosquitoes are the most common type of arthropods that spread diseases including malaria, dengue, West Nile, Zika, and chikungunya fevers. Mosquito-transmitted diseases affect hundreds of millions of people each year, resulting in about 750,000 deaths every year [22,23,24,25]. During a blood meal, an adult female mosquito punctures the skin of a vertebrate and ingests the fluid. If the vertebrate is infected, the virus is also ingested along with the blood. The virus can then replicate, cross the midgut barrier, and reside in the salivary gland of the mosquito at high titers [26,27,28,29]. During the subsequent blood meal, the contents of the salivary gland are released below the skin to counteract the host’s hemostasis and inflammatory responses, allowing the virus to enter the epidermis and dermis [30,31,32].

Mosquitoes’ innate immune responses allow them to survive infections, making them effective carriers. They use the RNA interference pathway for protection against viral infections, including arboviruses [33,34]. In addition, *Aedes aegypti* mosquitoes have been shown to use the JAK/STAT pathway in response to WNV, DENV, and YFV [35]. Some species of mosquitoes also utilize the immediate response of apoptosis in the salivary glands and midgut to control viral load [36,37].

The mosquito plays a major role in creating an ideal environment for virus entry and replication. Indeed, the vector secretes anti-hemostatic, angiogenic, and vasodilatory molecules through its saliva to maintain optimum blood flow during feeding [31,32,38,39]. This microenvironment allows for enhanced infection and pathogenicity of the virus by controlling the initial replication of the virus and the potential for the infection to become systemic [38,40,41,42,43,44,45]. While some groups have hypothesized suppression of anti-viral immune responses by mosquito saliva during infection, Pingen et al. (2016) showed that mosquito bites facilitate infection by triggering a cellular influx that is inadvertently beneficial for the virus [31,32,38,39,46].

The saliva of a mosquito has been shown to contain highly active molecules involved in modulating early viral infection. Indeed, while infecting mice with arboviruses through a mosquito bite or a needle accompanied by an uninfected mosquito bite, the former resulted in more severe disease [17,18,19,20,47]. Furthermore, in chickens infected with West Nile virus (WNV) via mosquito bite, significantly high viral titers were observed in the serum compared to the group infected using a needle [48]. Similar augmentation of viremia was seen in mice infected with WNV via bite accompanied by faster neuro-invasion compared to needle-inoculated animals [42]. 

There are many effects a mosquito bite has on immune cells [49]. Some species of mosquitoes, namely *Anopheles stephensi* and *Anopheles gambiae*, secrete saliva that can result in chemotactic activity. The vascular permeabilization and mast cell degranulation in the skin caused by the saliva were shown to recruit dendritic cells to the feeding site and neutrophils to the draining lymph node [50,51,52]. Another study using humanized mice showed that seven days post-mosquito bite, there was a decrease in IL-8, a neutrophil chemoattractant, in the serum correlating to a decrease in circulating neutrophils. This corresponded to increased neutrophils in the skin [49]. 

Pingen et al. used mice infected with aedine mosquito-borne Semliki Forest virus (SFV), an alphavirus shown to replicate efficiently in immune-competent mice, and Bunyamwera virus (BUNV), a genetically unrelated RNA virus [6,46,53,54]. In their study, mosquito bites induced an influx of inflammatory neutrophils, which, in turn, promoted myeloid cell entry into the bite site in a CCR2-dependent manner. This augmented viral infection since myeloid cells are permissive to the virus. Interestingly, viral infection via bite synergistically enhanced CXCL2 and IL-1β expression, and neutrophil influx compared to bite alone [46]. The researchers further confirmed the role of neutrophils by depleting them and blocking inflammasome activity. This resulted in decreased inflammation and a suppressive environment for the viral infection. Depleting neutrophils also significantly reduced edema by further enhancing the vascular leakage caused by the bite. In addition, neutrophil influx into the bite site at earlier stages of infection was required for the induction of vital bite-associated genes such as IL-1β, CCL2, CCL7, and CCL12. Importantly, neutrophil depletion did not affect virus-induced genes, while neutrophils expressing IL-1β were necessary for establishing cutaneous inflammatory responses to mosquito bites [46]. Therefore, factors secreted by the mosquito augment infection by increasing neutrophil-mediated inflammation at the bite site during early stages of infection, which later determines the systemic course of the infection in mice. At later stages, however, neutrophils were required to effectively resolve the infection and decrease mortality in mice. Indeed, higher number of the neutrophil-depleted mice infected with the more virulent SFV6 succumbed to infection compared to neutrophil-sufficient mice [46].

## 3. Zika Virus

Zika virus (ZIKV) is a flavivirus transmitted mainly by *Aedes* species mosquitoes. ZIKV generally causes fever, cutaneous rash, headache, and malaise [55,56]. However, in the most recent 2015–2016 epidemic in Latin America and the Caribbean that affected more than 1.5 million people [57], ZIKV caused severe congenital malformations in the fetus, commonly known as Congenital Zika Syndrome, [58,59] and Guillain-Barre syndrome [60,61].

Recently, Hastings et al. conducted a study to identify specific antigenic salivary gland proteins in the *Aedes aegypti* mosquito that promotes ZIKV pathogenesis [62]. They used yeast display to identify a molecule in the saliva of the mosquito that can activate neutrophils in the host. The authors named this previously undescribed protein as neutrophil-stimulating factor 1 (NeSt1). When mice were treated with NeSt1-blocking antibodies before being bitten by ZIKV-infected mosquitoes, they had a 50% higher survival rate compared to untreated mice. Furthermore, NeSt1 was shown to activate neutrophils inducing their expression of IL-1β, and monocyte/macrophage-attracting chemokines CXCL2 and CCL2. The recruited macrophages may then be infected by the virus increasing the viral load [62]. Overall, NeSt1 stimulated neutrophils at the bite site augmenting early viral infection and ZIKV pathogenesis (Figure 1).

In contrast, another study using adult AG129 interferon α/β receptor knockout mice infected with a recent strain of ZIKV showed the protective effects of neutrophils [63]. In this mouse model, ZIKV has been shown to infect astrocytes and neurons in the brain and spinal cord. Zukor et al. observed that this infection resulted in astrogliosis along with T cell and neutrophil infiltration. The neutrophil recruitment inversely correlated with the virus-induced paresis protecting infected mice from motor deficits, indicating that neutrophils may be required for controlling ZIKV-induced disease [63]. Mechanisms underlying this protection need to be further explored. It is important to note that the differences in neutrophil activity observed in this study compared to Hastings et al. may be attributed to the absence of a mosquito vector or mosquito salivary components during infection of mice.

## 4. Dengue Virus

Dengue virus (DENV) can cause clinical outcomes that range from mild febrile illness to dengue fever to dengue hemorrhagic fever to life-threatening dengue shock syndrome [64]. With approximately 2.5 billion people at risk globally, DENV is the most common arbovirus [65,66]. Clinical studies of adult dengue patients showed severe neutropenia with lowest levels occurring five days post-infection [67,68]. The neutropenia, however, was not predictive of severe virus-induced disease or associated with prolonged hospital stay or death [67]. Interestingly, the low level of neutrophils was not for a lack of activation signals. In fact, neutrophil-activating cytokines, IL-8 and TNF-α, were high during DENV infections [69], while neutrophil-associated genes such as DEF4A, CEACAM8, BPI, and ELA2 were upregulated in the blood during severe DENV infection [70]. Neutrophil elastase levels were also increased in DENV-infected patients compared to uninfected controls, with higher elastase activity in patients with dengue hemorrhagic fever compared to dengue fever patients [71]. This suggests that enhanced neutrophil activation can be associated with severe disease. 

In another study, researchers observed the formation of neutrophil extracellular traps (NETs) in vitro induced by DENV [72]. NET formation or NETosis consists of nuclear decondensation and delobulation, plasma membrane rupture, and release of DNA fibers that have anti-microbial peptides [73]. Although NETs play a crucial role when fighting infections, excessive NETosis and/or ineffective NET clearance can contribute to development of autoimmune diseases and inflammatory disorders [74,75]. Indeed, several NET-associated molecules, such as double-stranded DNA, histones, etc., are known to be autoantigens in systemic autoimmune diseases [74]. For instance, autoantibodies against NET components have been seen in systemic lupus erythematosus patients as well as an imbalance between NET formation and clearance, making them more prone to NET-mediated tissue damage [76,77,78,79]. Furthermore, NETs have also been implicated in the pathogenesis of inflammatory conditions including, but not limited to, small vessel vasculitis, psoriasis, and gout [74].

Examining the phenotypic and functional responses of neutrophils in adult dengue patients, Opasawatchai et al. observed an upregulation of CD66b on neutrophils and early stages of NET formation, indicating an activated state, during acute DENV infection [80]. CD66b is a granulocyte activation marker involved in degranulation and production of reactive oxygen species (ROS), which is essential for antiviral activity [81,82]. Interestingly, higher levels of NET components, IL-8, and TNF-α were found in patients diagnosed with the more severe dengue hemorrhagic fever compared to patients with dengue fever or healthy controls [80]. A study by Lien et al. identified the viral factor crucial for inducing NETosis in vitro and in mice to be DENV envelope protein domain III (EIII). This NET formation was alleviated in neutrophils from NLRP3 inflammasome-deficient mice, decreasing inflammation. Blocking EIII-neutrophil interactions also suppressed the NETosis [83].

The most severe disease caused by DENV comprises of systemic inflammation and increased vascular permeability. Many studies have also shown the activation of macrophages and platelets leading to an increase in proinflammatory cytokines and extracellular vesicles (EVs) [84,85,86,87] that transport proteins, peptides, and nucleic acids from one cell to another to modulate cell functions [88]. Indeed, DENV-induced release of IL-1β-containing EVs by platelets increased vascular permeability [87].

In addition, DENV enhanced release of EVs by activated platelets, which further activated CLEC5A, a spleen tyrosine kinase (Syk)-coupled C-type lectin receptor, and toll-like receptor 2 (TLR2) on neutrophils and macrophages. This induced NET formation and proinflammatory cytokine release [89]. Activation of CLEC5A is known to trigger NALP3 inflammasome activation and proinflammatory cytokine response [85,86,90], which augments systemic vascular permeability and hemorrhagic shock [86,91]. While blocking CLEC5A did not fully protect mice infected with a lethal dose of DENV [91], simultaneous blockade of CLEC5A and TLR2 significantly alleviated virus-induced inflammation and improved survival [89]. Together, these studies highlight the complex ways in which neutrophils mediate disease during the different stages of DENV infection.

## 5. West Nile Virus

Belonging to the same Flaviviridae family as DENV, West Nile virus (WNV) is a neuroinvasive pathogen [92]. WNV infection is typically only symptomatic in the elderly and immunocompromised individuals causing life-threatening neurological disease such as meningitis and encephalitis [92,93,94,95]. Strikingly, high levels of neutrophils were found in the cerebrospinal fluid collected from patients with WNV-induced disease, suggesting a major role of neutrophils in viral pathogenesis [96,97].

In mice infected with WNV, a rapid influx of neutrophils was seen at the site of infection promoting viral replication. Indeed, the expression of CXCL1 and CXCL2, neutrophil-attracting chemokines, was significantly upregulated in macrophages upon infection [5]. Interestingly, neutrophil-depletion studies revealed a dual role of these leukocytes during infection. Neutrophils were required for effective clearance of WNV and survival shown by higher viremia and death rate in mice depleted of neutrophils after infection. However, these cells were detrimental to the mice during early stages of infection since neutrophil depletion before WNV infection reduced viral burden and enhanced survival [5]. Overall, neutrophils can serve as reservoirs for WNV replication and dissemination as well as help defend against the virus at different stages of infection.

## 6. Alphaviruses

The alphavirus genus consists of many arthropod-borne viruses that are typically divided into two main groups, New World and Old World alphaviruses. New World alphaviruses such as eastern equine encephalitis virus (EEEV), western equine encephalitis virus (WEEV), and Venezuelan equine encephalitis virus (VEEV) cause encephalomyelitis in humans and are found in North and South America [98]. Old World alphaviruses that include chikungunya virus (CHIKV), Ross River virus (RRV), Mayaro virus, and o’nyong-nyong virus, are now found in Europe, Africa, Asia, and Oceania and generally induce fever, rash, and arthritis [8,99].

Although New World alphaviruses, such as EEEV, can have mortality rates as high as 70%, while Old World alphaviruses rarely cause death, the latter has caused many epidemics in the past, resulting in high infection rates [100]. A RRV epidemic in 1979–1980 in the South Pacific involved more than 60,000 patients [101] while the o’nyong nyong virus infected approximately 2 million people in Africa in the 1959–1962 epidemic [102]. CHIKV has caused reoccurring epidemics in numerous countries around the Indian Ocean since 2004 with millions of confirmed cases [103] and a surprising emergence in Europe and the Pacific Region for the first time in 2007 and 2011, respectively [104,105,106,107].

Humans and horses infected with New World alphaviruses show changes in the central nervous system characterized by high levels of neutrophil infiltration during early stages of disease, which is replaced by lymphocytes as the disease progresses [108,109]. Due to the lack of literature on the roles of neutrophils during New World alphavirus infection, we will focus on Old World alphaviruses in this section.

Old World alphaviruses can cause musculoskeletal inflammatory disease in humans that can be significantly debilitating. Infection with arthritis/myositis-associated alphaviruses can present with fever, joint pain, myalgia, and impaired movement [101,110]. Importantly, the musculoskeletal pain induced by arthritogenic alphaviruses can last for months to years in RRV- or CHIKV-infected individuals [111,112,113,114,115,116,117]. Many studies have been conducted to determine the cause of such chronic pain. In one study by Stoermer et al., RRV infection in mice with specific deletion of arginase 1 (Arg1) in neutrophils and macrophages was well controlled at later stages of infection enhancing viral clearance from musculoskeletal tissues and improving skeletal muscle tissue pathology [118]. Arg1 is expressed by monocytes/macrophages, neutrophils, and myeloid-derived suppressor cells (MDSCs) and plays an important role in regulating immune responses [119,120,121]. Although LysMCre Arg1^f/f^ mice, with conditional deletion of Arg1 in macrophages and neutrophils, had no change in the disease outcomes during the acute phase of infection, significantly enhanced protection was observed in the late stages of RRV infection [118]. Furthermore, conditional knockout of Arg1 substantially reduced Arg1 expression in musculoskeletal tissues following CHIKV and RRV infection, suggesting that macrophages and neutrophils are the predominant cells at the inflammatory sites following arthritogenic alphavirus infection [118]. Overall, the study highlighted the crucial role of Arg1 in contributing to disease severity. Specific neutrophil depleting methods such as Ly6G antibody treatments could help further narrow down the responsible cell type.

CHIKV infection is primarily characterized by macrophage and monocyte infiltration into the primary sites of virus replication, which are typically the skin, muscle, and joints. However, an influx of neutrophils, dendritic cells, natural killer cells, and lymphocytes has also been observed [122]. Indeed, resident cells at the site of infection produce neutrophil-attracting chemokines, CXCL1 and CXCL2, following other viral infections [123,124]. This chemokine production by resident cells remains to be seen during CHIKV infection. The recruited neutrophils produce ROS and other cytotoxic mediators to decrease viral replication [125]. In non-mammalian models of CHIKV infection such as zebrafish, the neutrophils also serve as an important source of type I interferon for eliminating the virus and alleviating disease [126]. Even in the absence of active viral replication during chronic phases of infection, CHIKV-induced arthritis may progress due to increased cytokine expression and immune cell infiltration [122,127]. 

A recent study found the role of CXCL10, a chemoattractant for monocytes/macrophages and T cells, during alphaviral infections using CHIKV and o’nyong nyong mouse models. At the peak of arthritic disease, which occurs 6 to 8 days post infection in mice, CXCL10^−/−^ mice had decreased levels of immune infiltration as well as viral loads at the site of viral inoculation, the footpad, compared to wild-type mice [128]. The predominant populations in the infiltrates were macrophages and neutrophils in the wild-type mice following infection but this influx was significantly reduced in the CXCL10^−/−^ mice. Interestingly, viral RNA was detected in these immune cells in wild-type mice, which was also significantly decreased in the knockout mice [128]. 

In another study, the role of NETs during CHIKV infection was explored. Ex vivo stimulation and infection of mouse-isolated neutrophils induced the release of NETs in a TLR7- and ROS-dependent manner neutralizing CHIKV [129]. The researchers used TLR3/7/9 triple knockout mice with TLR3^−/−^ and TLR9^−/−^ mice as controls due to the unavailability of TLR7^−/−^ mice. Although knockout of TLR3 and TLR9 did not affect NET production after CHIKV infection, there may be some synergistic effects of the triple knockout affecting virus-mediated NET release [129]. In vivo infection of IFNAR^−/−^ mice following NET inhibition enhanced susceptibility of the mice to an acute CHIKV infection confirming a crucial antiviral role of NETs. Moreover, clinical data also showed a correlation between the level of NETs in the blood and systemic viral loads in CHIKV infected patients [129,130,131]. Even though the role of NETs has been established during an acute CHIKV infection, they may also play a part during chronic infection. Indeed, neutrophils infiltrate the synovium and release NETs leading to damage of the joint tissues in rheumatoid arthritis [132]. 

CCR2 has been implicated in playing a protective role during CHIKV infection by preventing neutrophil-mediated pathology. CCR2^−/−^ mice infected with CHIKV in the hind feet showed decreased levels of monocyte/macrophage infiltration with substantial increase in neutrophil infiltration, followed by eosinophils, compared to wild-type mice [133]. This change in cellular influx was associated with increased levels of CXCL1, CXCL2, G-CSF, and IL-1β with a decrease in IL-10, promoting neutrophil recruitment and exacerbating inflammation [134,135,136,137,138,139,140]. The eosinophil infiltration may be promoted by neutrophil-induced tissue damage to help control the inflammation in infected CCR2^−/−^ mice [141]. CCR2 deficiency also led to cartilage damage in mice following CHIKV infection, which is normally not a symptom of alphaviral arthritis [133]. In fact, elevated macrophage and neutrophil infiltrates in CCR2^−/−^ mice with collagen-induced arthritis is accompanied by more severe disease [134,142]. Interestingly, Poo et al. attempted neutrophil depletion in CCR2^−/−^ mice after CHIKV infection, which resulted in new pathology characterized by increased foot swelling along with widespread hemorrhage and edema [133,143]. This, once again, may be suggestive of a dual role of neutrophils, where they are detrimental during certain stages of infection while protective during others. 

Another group delineated the role of neutrophils during a pathogenic CHIKV infection on B cell maturation and lymphocyte influx. McCarthy et al. found that mice infected with a wild-type, not acutely cleared, strain of CHIKV had recruitment of monocytes and neutrophils to the draining lymph node (dLN). This aberrantly affected lymphocyte accumulation, lymph node organization, and virus-specific B cell responses, which was reversed by blocking the influx [144]. Interestingly, only pathogenic CHIKV decreased germinal center formation in the dLN, resulting in lower neutralizing antibodies in the serum compared to infection with an attenuated strain [145,146]. These diminished B cell responses were improved upon depletion of monocytes and neutrophils during early stages of infection [144].

Depleting either monocytes or neutrophils did not restore lymphocyte counts in the dLN, indicating that one of the two cell types is sufficient to block lymphocyte infiltration [144]. Furthermore, mice lacking type I interferon signaling (IFNAR^−/−^) had higher percentage of neutrophils in the dLN compared to wild-type mice following pathogenic CHIKV infection. In contrast, MyD88-deficient mice and wild-type mice treated with IL-1 receptor (IL-1R) blocking antibody at the time of infection had reduced the percentage of neutrophils [144]. Together, MyD88-IL-1R signaling plays a crucial role in promoting the accumulation of neutrophils in the dLN while type I interferon signaling inhibits the recruitment during pathogenic CHIKV infection [144].

While IFN-α was observed to inhibit neutrophil influx into the dLN, IFN-β was found to inhibit neutrophil infiltration into the musculoskeletal tissues during CHIKV infection [147]. Following CHIKV inoculation in the foot of IFN-β^−/−^ mice, although no change was observed in viral load in the foot or the blood compared to wild-type mice, there were increased levels of neutrophils in the foot [147]. Neutrophil depletion in IFN- β^−/−^ mice alleviated musculoskeletal disease induced by CHIKV observed through significantly reduced foot swelling. On the other hand, IFN-α^−/−^ mice had higher viral burdens at the site of infection and in circulation [147]. This indicates that IFN-α helps limit viral replication whereas IFN-β modulates neutrophil recruitment to the site of infection that is necessary for exacerbation of disease pathology (Figure 2). Curiously, neither neutrophil-attracting chemokines nor proinflammatory cytokines were upregulated in the IFN-β^−/−^ mice to accompany the neutrophil-mediated inflammation making the mechanism through which IFN-β regulates neutrophil infiltration during acute CHIKV infection unclear [147].

It is important to note that most studies deplete neutrophils in vivo to understand their function. All the studies involving neutrophil depletion referenced in this review use Ly6G antibody treatments in mice. While Ly6G may be transiently expressed on many myeloid cells in the bone marrow including monocytes and other granulocytes, neutrophils that are circulating and recruited to the site of inflammation typically have higher Ly6G expression [148]. Basophils and eosinophils are thought to be Ly6G^-^ or Ly6G^low/intermediate^ [148]. Additionally, some studies showed that Ly6G-mediated neutrophil depletion reduced only the Ly6C^intermediate^ neutrophil population and not the Ly6C^high^ monocyte population [144].

## 7. Conclusions

Neutrophils are key players in the immune system, being the most abundant leukocytes. They are one of the first responders to the site of infection. However, the heterogeneity of their roles and the variability from one infection to another makes it difficult to determine if the effects will be beneficial or detrimental to the host (Table 1). Arboviruses not only induce neutrophil-mediated inflammation using viral factors but also through factors in their vector, adding another level of complexity. Their mode of transmission through arthropods immensely increases the rate at which they spread, highlighting the need for better understanding of the underlying mechanisms involved in pathogenesis. During arboviral infections, the time and amount of neutrophil infiltration to the site of infection may have a significant effect on the outcome. Following infection, an early influx with a high number of hyperactivated neutrophils releasing high levels of IL-1β, ROS, and NETs may augment infection and disease. However, an influx at later stages of infection may be protective. Regularly causing epidemics in the vulnerable areas of the world, arboviral infections need to be controlled with unique therapeutics that can control the vector-mediated and virus-mediated symptoms. Neutrophils are implicated in disease pathology induced by arboviruses and their vectors, making them a potential therapeutic target. In-depth understanding of the neutrophil pathways involved may be crucial for successful treatment of arboviral infections.

## Figures and Tables

**Figure 1 cells-10-01324-f001:**
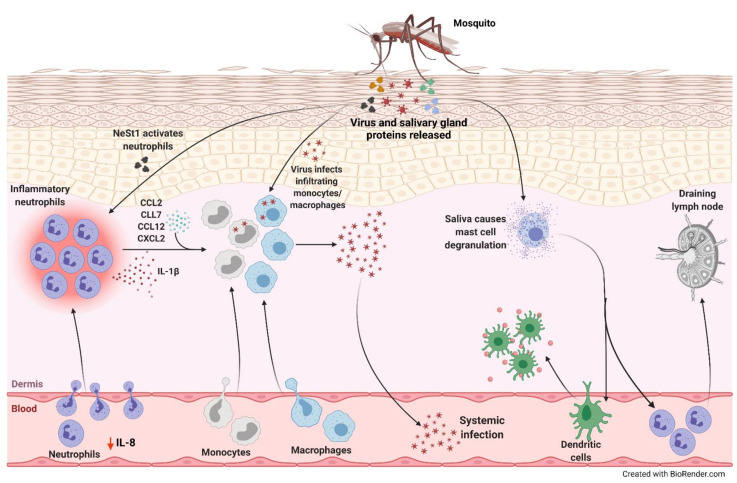
Neutrophil-mediated viral replication and dissemination induced by mosquito saliva. During a blood meal, mosquito carrying an arbovirus injects the virus along with its salivary gland proteins below the skin of the host. There is a decrease in IL-8 levels in the serum correlating to lower number of circulating neutrophils and higher number in the skin. One of the proteins in the saliva, neutrophil-stimulating factor 1 (NeSt1), activates the neutrophils in the dermis, the deepest layer of the skin, which houses the immune cells. IL-1β is secreted by these inflammatory neutrophils to establish cutaneous response to the bite. Additionally, bite-associated monocyte/macrophage-attracting chemokines, CCL2, CCL7, CCL12, and CXCL2, are upregulated. The infiltrating monocytes and macrophages are permissive to infection enhancing viral replication and increasing the potential for systemic spread. The mosquito saliva also causes vascular permeabilization and mast cell degranulation in the skin recruiting dendritic cells to the bite site, contributing to the inflammation, and neutrophils to the draining lymph nodes.

**Figure 2 cells-10-01324-f002:**
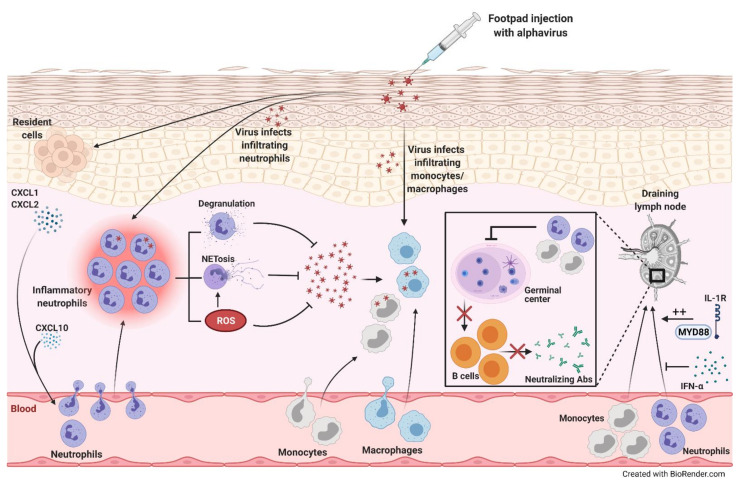
Alphavirus infection-induced neutrophil recruitment and inflammation. Following footpad injection of mice with alphavirus, levels of neutrophil-attracting chemokines, CXCL1 and CXCL2, increase. These chemokines and CXCL10 recruit neutrophils, which release reactive oxygen species (ROS), neutrophil extracellular traps (NETs), and other cytotoxic mediators through degranulation, promoting viral clearance. The infiltrating neutrophils can also be directly infected, triggering NET release in a ROS-dependent manner. Furthermore, monocytes/macrophages infiltrating the site of infection can be directly infected. On the other hand, alphaviral infections induce monocyte and neutrophil recruitment into the draining lymph node (dLN) that inhibit germinal center formation decreasing B cell maturation and neutralizing antibody (Ab) production. MyD88-IL-1R signaling promotes the accumulation of neutrophils in the dLN, while IFN-α inhibits this influx.

**Table 1 cells-10-01324-t001:** Summary of the roles of neutrophils during various arboviral infections.

Arbovirus	Beneficial Roles of Neutrophils	Detrimental Roles of Neutrophils	Replication in Neutrophils
Zika virus (ZIKV)	• In AG129 IFN-α/β receptor knockout mice, ZIKV induced neutrophil infiltration, which inversely correlated with virus-induced paresis protecting mice from motor deficits [63]	• Neutrophil-stimulating factor 1 (NeSt1) in mosquito saliva activated neutrophils, inducing IL-1β, CXCL2 and CCL2 expression, which augmented early viral infection in mice [62]	
Dengue virus (DENV)		Clinical studies: Neutrophil-associated genes such as DEF4A, CEACAM8, BPI, and ELA2 were upregulated in the blood during severe DENV infection [70] Patients with the more severe dengue hemorrhagic fever had higher neutrophil elastase activity [71] and higher levels of NET components [80] than dengue fever patients or healthy controls In adult patients with acute DENV, CD66b was upregulated on neutrophils and there were early stages of NET formation [80] Mice: DENV enhanced release of extracellular vesicles by activated platelets activating CLEC5A and TLR2 on neutrophils and macrophages inducing NET formation and proinflammatory cytokine release decreasing survival [89]	
West Nile virus (WNV)	• Neutrophil depletion after infection increased viremia and death rate in mice [5]	• Neutrophil depletion before infection reduced viral burden and enhanced survival in mice [5]	Yes [5]
Semliki Forest virus (SFV)	Mice: • At later stages of infection, neutrophils were required to resolve the infection and decrease mortality [46]	Mice: Mosquito bite induced an influx of inflammatory neutrophils which promoted CCR2-dependent myeloid cell entry augmenting viral infection [46] Mosquito bite and virus synergistically enhanced CXCL2 and IL-1β expression, and neutrophil influx [46] Depleting neutrophils and blocking inflammasome activity in the early stages of infection decreased inflammation, suppressed viral infection, and reduced edema [46]	
Ross River virus (RRV)		• Deletion of Arg1 in macrophages and neutrophils enhanced viral clearance and improved skeletal muscle tissue pathology in late stages of infection in mice, with no effect in the acute phase of infection [118]	
Chikungunya virus (CHIKV)	Zebrafish: • Neutrophils served as a major source of type I interferon for eliminating the virus and alleviating disease [126] Mice: Neutrophils induced release of NETs in a TLR- and ROS-dependent manner, neutralizing CHIKV during an acute infection [129] Neutrophil depletion in CCR2^−/−^ mice after CHIKV infection increased foot swelling along with widespread hemorrhage and edema [133]	Mice: CCR2 deficiency increased neutrophil infiltration at the site of infection exacerbating inflammation [133] Recruitment of neutrophils to the draining lymph node (dLN) following pathogenic CHIKV infection affected lymphocyte accumulation, lymph node organization, and decreased germinal center formation resulting in lower virus-specific neutralizing antibodies in the serum [144] Neutrophil depletion in IFN-β^−/−^ mice alleviated CHIKV-induced musculoskeletal disease with reduced foot swelling [147]	
O’nyong nyong virus (ONNV)		• CXCL10^−/−^ mice had reduced influx of macrophages and neutrophils, which was associated with decreased viral loads and foot swelling [128]	Yes [128]

## Data Availability

Not applicable.

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
