# Peer review of "Complex Roles of Neutrophils during Arboviral Infections"

_cells, 2021, doi:10.3390/cells10061324_

Round 1

Reviewer 1 Report

This review summarises some of the key aspects involved with neutrophil responses to arbovirus infection. Congratulations to the authors for putting such a comprehensive body of literature together in one review. There are however quite a few inaccuracies that should be corrected prior to publication and are listed below.

There are a number of references that have been used that do not study arboviruses – please remove these or make it clear in the text which viruses have been used.

Page 1

  • Last line of paragraph one. There is no evidence in these papers cited (ref 4,5) that show WNV can infect neutrophils. Neither can this reviewer recall reading any other such paper. This should be corrected to state that neutrophils are generally refractory to infection with most arboviruses (that is not to say that neutrophil responses aren’t key). This error is repeated on page 5 – please remove any statement that WNV can infect neutrophils.
  • Note that viruses should only have capital letter if they are named after a location or person e.g. “Chikungunya virus” should be ‘chikungunya virus’. Please correct e.g. in paragraph 2, page 1.

Page 2

  • Page two, the authors state “Ticks, biting flies and mosquitoes enhance disease severity as well as transmission of many viruses [17–22].” However, there is no evidence that factors from biting flies enhance infection with arbovirus. Instead, the reference 22 refers to the effect of sand fly biting to infection with a parasite, which is quite distinct to virus infection. This should be corrected.
  • Reference 31 does not seem relevant to this point “During the subsequent blood meal, the contents of the salivary gland are released below the skin to counteract the host’s hemostasis and inflammatory responses allowing the virus to enter the epidermis and dermis [31].” Please provide an appropriate reference.
  • Re: Paragraph starting “The mosquito plays…”. While mosquito salivary proteins have many roles in vertebrates, there is no convincing data to show they are “anti-inflammatory”. Indeed, salivary and biting is highly inflammatory. Please remove this description. More recent data in reference 48 comprehensively demonstrates that mosquito biting does not suppress induction of most immune responses to virus.   
  • When discussing host response to Anopheles biting/saliva, it should be made clear that these vectors do not transmit arboviruses and therefore may not be relevant.
  • Pingen et al study also made use of the genetically distinct Bunyamwera virus (in addition to SFV) to define role of host inflammatory response to mosquito biting. This should be corrected.

Page 3

  • In reference 58 they do not show infection of macrophages with ZIKV – this should be removed.
  • Importantly in reference 59 they do not infect mice via mosquito or with mosquito saliva. Therefore, comparisons to role of neutrophils in reference 58 are difficult when assessing the role of neutrophils. This should be clarified.

Page 6

  • Paragraph starting “CHIKV infection is primarily characterized…” should be completely re-written. Importantly most of the refences are completely wrong – referencing other viruses. The one reference with CHIKV (ref 113) is concerning infection of fish, which are not a natural host for CHIKV infection. Importantly, it should be made clear that neutrophils are not known to be major source of type I IFNs in mammals – the zebrafish is an exception and may be due to its non-mammalian status.
  • Last paragraph. CXCL10 is not chemoattractive to macrophages or monocytes, which lack CXCR3 expression. CXCL10 attracts CXCR3 bearing cells such as Th1 and CD8 T cells. Any decrease in monocyte recruitment, e.g. as seen in cxcl10-decicient mice in reference 116, will be an indirect effect of reduced recruitment of these cells.

Page 7

  • Reference 135 is from 1980 and used crude homogenates (including type I and type II IFNs) – therefore it is not possible to discern any role for specific IFNs from this study – not do they study arboviruses. I would suggest not discussing this work. The last two paragraphs should be re-worded accordingly. Indeed, it is also not clear which reference is being referred to in many of these statements – please clarify or  

Page 8

  • There is no evidence that alphavirus arboviruses infect neutrophils. Please remove this from figure 2 and any references in the text.
  • There is no evidence that neutrophils make type I IFNs. Please provide appropriate references for this – as this reviewer can’t find any in the literature to support this statement, or simply remove this from figure 2 and any reference in the text.

Author Response

This review summarises some of the key aspects involved with neutrophil responses to arbovirus infection. Congratulations to the authors for putting such a comprehensive body of literature together in one review. There are however quite a few inaccuracies that should be corrected prior to publication and are listed below.

There are a number of references that have been used that do not study arboviruses – please remove these or make it clear in the text which viruses have been used.

Page 1

  • Last line of paragraph one. There is no evidence in these papers cited (ref 4,5) that show WNV can infect neutrophils. Neither can this reviewer recall reading any other such paper. This should be corrected to state that neutrophils are generally refractory to infection with most arboviruses (that is not to say that neutrophil responses aren’t key). This error is repeated on page 5 – please remove any statement that WNV can infect neutrophils.

Response:

Thank you for the comment. We agree with the reviewer that there are not many studies showing arboviral infection of neutrophils. However, Bai et al. (ref 5) showed that neutrophils are an important reservoir for West Nile virus. Ex vivo infection of murine and human neutrophils with WNV was significantly higher than infection of macrophages. Similarly, PMNs isolated from the blood of WNV-infected mice 3 days post-infection had ~8-fold higher viral load than the other cell types combined.  

  • Note that viruses should only have capital letter if they are named after a location or person e.g. “Chikungunya virus” should be ‘chikungunya virus’. Please correct e.g. in paragraph 2, page 1.

Response:

We apologize for this typo. The names of the viruses have been corrected accordingly throughout the manuscript.

Page 2

  • Page two, the authors state “Ticks, biting flies and mosquitoes enhance disease severity as well as transmission of many viruses [17–22].” However, there is no evidence that factors from biting flies enhance infection with arbovirus. Instead, the reference 22 refers to the effect of sand fly biting to infection with a parasite, which is quite distinct to virus infection. This should be corrected.

Response:

Thank you for the comment. We have removed ref 22 and that sentence on Page 2 now reads:

“Ticks and mosquitoes enhance disease severity as well as transmission of many viruses [17-21].”

  • Reference 31 does not seem relevant to this point “During the subsequent blood meal, the contents of the salivary gland are released below the skin to counteract the host’s hemostasis and inflammatory responses allowing the virus to enter the epidermis and dermis [31].” Please provide an appropriate reference.

Response:

We apologize for this typo. Ref 31 and 32 (ref 33 and 34 in the original manuscript) have been added to the end of that sentence on Page 2.

  • Re: Paragraph starting “The mosquito plays…”. While mosquito salivary proteins have many roles in vertebrates, there is no convincing data to show they are “anti-inflammatory”. Indeed, salivary and biting is highly inflammatory. Please remove this description. More recent data in reference 48 comprehensively demonstrates that mosquito biting does not suppress induction of most immune responses to virus.   

Response:

We agree with the reviewer that ref 48 (ref 46 in the revised manuscript) shows alterations to the immune responses in the presence of mosquito saliva instead of suppression of anti-viral responses. As such, this section on Page 2 now reads:

“While some groups have hypothesized suppression of anti-viral immune responses by mosquito saliva during infection, Pingen et al. (2016) showed that mosquito bites facilitate infection by triggering a cellular influx that is inadvertently beneficial for the virus [31,32,38,39,46].”  

  • When discussing host response to Anopheles biting/saliva, it should be made clear that these vectors do not transmit arboviruses and therefore may not be relevant.

Response:

A recent paper (Anopheles mosquitoes may drive invasion and transmission of Mayaro virus across geographically diverse regions (nih.gov)) investigating six mosquito species for their ability to support Mayaro virus (MAYV), an alphavirus, infection and transmission showed that all Anopheles species were able to transmit MAYV. Although the mosquito species used were laboratory strains, the diversity in the geographic origin of the species (Africa, Asia and North America) highlights the potential for Anopheles mosquitoes to be effective vectors for arboviruses.

  • Pingen et al study also made use of the genetically distinct Bunyamwera virus (in addition to SFV) to define role of host inflammatory response to mosquito biting. This should be corrected.

Response:

This section on Page 2 now reads:

“Pingen et al. used mice infected with aedine mosquito-borne Semliki Forest virus (SFV), an alphavirus shown to replicate efficiently in immune-competent mice, and Bunyamwera virus (BUNV), a genetically unrelated RNA virus [6,46,53,54].”

Page 3

  • In reference 58 they do not show infection of macrophages with ZIKV – this should be removed.

Response:

We apologize for the typo. This sentence on Page 3 now reads:

“The recruited macrophages may then be infected by the virus increasing the viral load [62].”

  • Importantly in reference 59 they do not infect mice via mosquito or with mosquito saliva. Therefore, comparisons to role of neutrophils in reference 58 are difficult when assessing the role of neutrophils. This should be clarified.

Response:

We agree with the reviewer. The following was added to Page 3:

“It is important to note that the differences in neutrophil activity observed in this study compared to Hastings et al. may be attributed to the absence of a mosquito vector or mosquito salivary components during infection of mice.”

Page 6

  • Paragraph starting “CHIKV infection is primarily characterized…” should be completely re-written. Importantly most of the refences are completely wrong – referencing other viruses. The one reference with CHIKV (ref 113) is concerning infection of fish, which are not a natural host for CHIKV infection. Importantly, it should be made clear that neutrophils are not known to be major source of type I IFNs in mammals – the zebrafish is an exception and may be due to its non-mammalian status.

Response:

We thank the reviewer for the comment. We agree that ref 111 and 112 (ref 123 and 124 in the revised Ms) in the original manuscript reference other viruses and we apologize for not clarifying that. We have also clarified that ref 113 (ref 126 in the revised Ms) references CHIKV infection in zebrafish. We agree with the reviewer about type I IFNs and have clarified this in the revised manuscript. In Figure 2, we have removed type I IFN production by neutrophils, and CXCL1 and CXCL2 production by resident cells.

This section on Page 6 now reads:

“CHIKV infection is primarily characterized by macrophage and monocyte infiltration into the primary sites of virus replication, which are typically the skin, muscle, and joints. However, an influx of neutrophils, dendritic cells, natural killer cells, and lymphocytes has also been observed [122]. Indeed, resident cells at the site of infection produce neutrophil-attracting chemokines, CXCL1 and CXCL2, following other viral infections [123,124]. This chemokine production by resident cells remains to be seen during CHIKV infection. The recruited neutrophils produce ROS and other cytotoxic mediators to decrease viral replication [125]. In non-mammalian models of CHIKV infection such as zebrafish, the neutrophils also serve as an important source of type I interferon for eliminating the virus and alleviating disease [126]. Even in the absence of active viral replication during chronic phases of infection, CHIKV-induced arthritis may progress due to increased cytokine expression and immune cell infiltration [122,127].”

  • Last paragraph. CXCL10 is not chemoattractive to macrophages or monocytes, which lack CXCR3 expression. CXCL10 attracts CXCR3 bearing cells such as Th1 and CD8 T cells. Any decrease in monocyte recruitment, e.g. as seen in cxcl10-decicient mice in reference 116, will be an indirect effect of reduced recruitment of these cells.

Response:

Thank you for the comment. We agree that CXCL10 mainly attracts CXCR3 bearing cells. However, there are reports showing CXCL10-induced recruitment of monocytes and macrophages (Inflammatory Cytokine Expression Is Associated with Chikungunya Virus Resolution and Symptom Severity (nih.gov); CXCL10 induces the recruitment of monocyte-derived macrophages into kidney, which aggravate puromycin aminonucleoside nephrosis (nih.gov)). Furthermore, in ref 116 in the original manuscript (ref 128 in the revised Ms), the authors mention that the number of T cells recruited to the mouse footpad following ONNV infection was comparable between WT and CXCL10-deficient mice.

Page 7

  • Reference 135 is from 1980 and used crude homogenates (including type I and type II IFNs) – therefore it is not possible to discern any role for specific IFNs from this study – not do they study arboviruses. I would suggest not discussing this work. The last two paragraphs should be re-worded accordingly. Indeed, it is also not clear which reference is being referred to in many of these statements – please clarify or  

Response:

We agree with the reviewer that ref 135 in the original manuscript is outdated. As such, we have removed that reference along with the statements derived from that paper. We apologize for the lack of clarity in the referencing in Page 7 and we have specified the references after the statements.

Page 8

  • There is no evidence that alphavirus arboviruses infect neutrophils. Please remove this from figure 2 and any references in the text.

Response:

Thank you for the comment. We agree with the reviewer that there are not many studies showing alphaviral infection of neutrophils. However, Lin et al. (ref 128 in the revised Ms) isolated different cell types from mice infected with o’nyong nyong virus 6 days post-infection and found the highest levels of viral RNA in neutrophils, with second highest levels in macrophages.    

  • There is no evidence that neutrophils make type I IFNs. Please provide appropriate references for this – as this reviewer can’t find any in the literature to support this statement, or simply remove this from figure 2 and any reference in the text.

Response:

We agree with the reviewer. Type I IFN production by neutrophils was seen in non-mammalian models of CHIKV infection such as zebrafish and has not been observed in mammals. As such, we have clarified this in the revised manuscript and removed this from Figure 2.

Reviewer 2 Report

It is a well-written manuscript and highly interesting and relevant.

However, it requires some major revision since highly relevant information is missing:

  1. The authors start with a chapter on mosquitos. What about the immune response in mosquitos? The authors should add a few sentences.
  2. Very often the authors refere to publications where neutrophils have been depleted to study neutrophil functions (e.g. ref. 48, ref. 5 West Nile). The authors should comment on the techniques used to deplete neutrophils, since often it is not neutrophil-specific depletion e.g. ly6G also stains inflammatory monocytes.
  3. In chapter 4 the authors introduce NETs, which is quite an important mechanism how neutrophils contribute to viral infections. The authors mention that NETs may have detrimental effects e.g. autoimmune diseases. This is by far not all. The authors should add more information. Furthermore, the literature on the role of NETs during Dengue infections is not complete. High-ranking publications on that topic are missing.
  4. In the introduction the authors mention that some viruses can replicate within the neutrophils. However detailed information on this phenomenon is missing throughout the text.
  5. The review is missing mechanistical details e.g. in case of Zika-virus the authors only mention that neutrophils may be required for controlling ZIKV-induced disease. However mechanistical details on the neutrophil functions are completely missing.
  6. I suggest to include a Table with an overview of the role of neutrophils including information on mechanisms, consequences etc. with 3 headlines to differentiate (1) detrimental consequences, (2) protective consequences and (3) replication in neutrophils.

Author Response

It is a well-written manuscript and highly interesting and relevant.

However, it requires some major revision since highly relevant information is missing:

  1. The authors start with a chapter on mosquitos. What about the immune response in mosquitos? The authors should add a few sentences.

Response:

Thank you for the comment. We have added the following to the revised manuscript:

“Mosquitoes’ innate immune responses allow them to survive infections making them effective carriers. They use the RNA interference pathway for protection against viral infections, including arboviruses [33,34]. In addition, Aedes aegypti mosquitoes have been shown to use the JAK/STAT pathway in response to WNV, DENV, and YFV [35]. Some species of mosquitoes also utilize the immediate response of apoptosis in the salivary glands and midgut to control viral load [36,37].”   

  1. Very often the authors refere to publications where neutrophils have been depleted to study neutrophil functions (e.g. ref. 48, ref. 5 West Nile). The authors should comment on the techniques used to deplete neutrophils, since often it is not neutrophil-specific depletion e.g. ly6G also stains inflammatory monocytes.

Response:

All studies referred to in this review that deplete neutrophils use Ly6G antibody treatments. While Ly6G may be transiently expressed on monocytes in the bone marrow, Ly6G antibody treatments are thought to be neutrophil-specific (Ly6 family proteins in neutrophil biology - Lee - 2013 - Journal of Leukocyte Biology - Wiley Online Library). Furthermore, ref 132 (ref 144 in the revised Ms) showed that Ly6G-mediated depletion did not affect the Ly6Chigh monocyte population but depleted the Ly6Cintermediate neutrophil population.

  1. In chapter 4 the authors introduce NETs, which is quite an important mechanism how neutrophils contribute to viral infections. The authors mention that NETs may have detrimental effects e.g. autoimmune diseases. This is by far not all. The authors should add more information. Furthermore, the literature on the role of NETs during Dengue infections is not complete. High-ranking publications on that topic are missing.

Response:

We agree with the reviewer and we have added more information about the detrimental effects of NETs. We apologize for missing high-ranking publications and we have included them in the revised manuscript. The following has been added to section 4:

“Although NETs play a crucial role when fighting infections, excessive NETosis and/or ineffective NET clearance can contribute to development of autoimmune diseases and inflammatory disorders [74,75]. Indeed, several NET-associated molecules, such as double-stranded DNA, histones etc., are known to be autoantigens in systemic autoimmune diseases [74]. For instance, autoantibodies against NET components have been seen in systemic lupus erythematosus patients as well as an imbalance between NET formation and clearance, making them more prone to NET-mediated tissue damage [76-79]. Furthermore, NETs have also been implicated in the pathogenesis of inflammatory conditions including, but not limited to, small vessel vasculitis, psoriasis, and gout [74].”

“A study by Lien et al. identified the viral factor crucial for inducing NETosis in vitro and in mice to be DENV envelope protein domain III (EIII). This NET formation was alleviated in neutrophils from NLRP3 inflammasome-deficient mice decreasing inflammation. Blocking EIII-neutrophil interactions also suppressed the NETosis [83].”

  1. In the introduction the authors mention that some viruses can replicate within the neutrophils. However detailed information on this phenomenon is missing throughout the text.

Response:

We thank the reviewer for the comment. Unfortunately, we were not able to find many studies exploring the potential of neutrophils to be reservoirs for viruses. Few of the studies that show effective infection and replication of viruses, such as WNV and ONNV, in neutrophils do not provide more details for the phenomenon. That sentence in the introduction now reads:

“In addition, some viruses such as influenza, specifically H5N1, and West Nile virus are known to infect and replicate within neutrophils, using these cells as reservoirs for dissemination, although mechanisms involved in this phenomenon remain unclear [4,5].”

  1. The review is missing mechanistical details e.g. in case of Zika-virus the authors only mention that neutrophils may be required for controlling ZIKV-induced disease. However mechanistical details on the neutrophil functions are completely missing.

Response:

Unfortunately, we were not able to find studies exploring the mechanisms underlying neutrophil-mediated control of ZIKV. As such, the end of the ‘Zika Virus’ section now reads:

“The neutrophil recruitment inversely correlated with the virus-induced paresis protecting infected mice from motor deficits, indicating that neutrophils may be required for controlling ZIKV-induced disease [63]. Mechanisms underlying this protection need to be further explored.”

  1. I suggest to include a Table with an overview of the role of neutrophils including information on mechanisms, consequences etc. with 3 headlines to differentiate (1) detrimental consequences, (2) protective consequences and (3) replication in neutrophils.

Response:

We agree with the reviewer that a table would improve this review. Table 1 in the revised manuscript summarizes the differences in the role of neutrophils during various arboviral infections.

Reviewer 3 Report

This review by Muralidharan and Reid details the current understanding of neutrophil involvement in arthropod-borne virus infection and pathogenesis.  Arthropod-borne viruses cause many significant diseases throughout the world and they will continue to be an important area of virology research.  A better understanding of virus/host interactions are key to developing measures of prevention and treatment.  The influence of the arthropod vector is also a key factor in the transmission and pathogenesis of these important pathogens.  The authors have effectively covered these layers of complexity and the review strikes a good balance of these topics.  The text is written well and the figures are clear.  It was a pleasure to read and I think it will provide a valuable resource to the field.    I have only a few suggestions, points that need clarification, and minor corrections listed below.

Comments:

  1. Would it be useful to include a table listing the relevant/key arboviruses covered in the review and the general findings as to whether neutrophil responses are beneficial, detrimental or both to the host? In general, a summary table can be a useful resource for investigators/students wanting a bird’s-eye view of the subject. It would not have to be overly complicated or detailed, but inclusion of key experimental findings (for example, in mouse models) versus clinical findings in humans might be of interest. With findings to date, how do the arboviruses compare or contrast with each other in relation to the role of neutrophils?  Does the current data suggest any common themes?

  1. Mosquito-transmitted “diseases infect” hundreds of millions… Should be changed to either “viruses infect” or “diseases affect”.

  1. Page 2, paragraph 4. This paragraph describes that mosquito bites have many effects on immune cells. It appears that both mouse and human data are referred to in this paragraph, but it is not always clear what statement/information is from experimental or clinical findings.  It will strengthen the review to clearly indicate human data, more specifically. 

  1. CD66b is a marker of neutrophil activation, but possibly expand on the general functional properties of CD66b expressing neutrophils. There will likely be many arbovirologists reading this review and may not be completely aware of general functions of activated neutrophils. Specifically, what is the potential significance of CD66b positive neutrophils in the context of these infections.

  1. Although Old World alphavirus infections can be associated with arthritis, the paragraph describing CHIKV infection (page 6) should include some details as to where in the body inflammatory infiltrates are seen. Again, some readers may not be completely familiar with the diseases associated with alphavirus infection. To step that out with more detail again might improve the review.

  1. Same paragraph on CHIKV, references 111 and 112 do not appear to be related to CHIKV.

  1. Page 6, first paragraph, last sentence needs editing. Should the second “with” be “and”?

  1. The figures are very appealing, but possibly increase font size or bold the text.

Author Response

This review by Muralidharan and Reid details the current understanding of neutrophil involvement in arthropod-borne virus infection and pathogenesis.  Arthropod-borne viruses cause many significant diseases throughout the world and they will continue to be an important area of virology research.  A better understanding of virus/host interactions are key to developing measures of prevention and treatment.  The influence of the arthropod vector is also a key factor in the transmission and pathogenesis of these important pathogens.  The authors have effectively covered these layers of complexity and the review strikes a good balance of these topics.  The text is written well and the figures are clear.  It was a pleasure to read and I think it will provide a valuable resource to the field.    I have only a few suggestions, points that need clarification, and minor corrections listed below.

Comments:

  1. Would it be useful to include a table listing the relevant/key arboviruses covered in the review and the general findings as to whether neutrophil responses are beneficial, detrimental or both to the host? In general, a summary table can be a useful resource for investigators/students wanting a bird’s-eye view of the subject. It would not have to be overly complicated or detailed, but inclusion of key experimental findings (for example, in mouse models) versus clinical findings in humans might be of interest. With findings to date, how do the arboviruses compare or contrast with each other in relation to the role of neutrophils?  Does the current data suggest any common themes?

Response:

We agree with the reviewer that a table would improve this review. Table 1 in the revised manuscript summarizes the differences in the roles of neutrophils during various arboviral infections. We have also added the following to the ‘Conclusion’ section:

“During arboviral infections, the time and amount of neutrophil infiltration to the site of infection may have a significant effect on the outcome. Following infection, an early influx with a high number of hyperactivated neutrophils releasing high levels of IL-1β, ROS and NETs may augment infection and disease. However, an influx at later stages of infection may be protective.”    

  1. Mosquito-transmitted “diseases infect” hundreds of millions… Should be changed to either “viruses infect” or “diseases affect”.

Response:

Thank you, this sentence has been changed to:

“Mosquito-transmitted diseases affect hundreds of millions of people each year resulting in about 750,000 deaths every year [22-25].”

  1. Page 2, paragraph 4. This paragraph describes that mosquito bites have many effects on immune cells. It appears that both mouse and human data are referred to in this paragraph, but it is not always clear what statement/information is from experimental or clinical findings.  It will strengthen the review to clearly indicate human data, more specifically. 

Response:

Ref 44 (ref 49 in the revised Ms) uses mice engrafted with human hematopoietic stem cells to determine effects of mosquito bites on human immune cells while ref 45-47 (ref 50-52 in the revised Ms) study mosquito saliva itself or the effects in mice. We clarified this in the manuscript and this section now reads:

“Another study using humanized mice showed that 7 days post-mosquito bite, there was a decrease in IL-8, a neutrophil chemoattractant, in the serum correlating to a decrease in circulating neutrophils. This corresponded to increased neutrophils in the skin [49].”

  1. CD66b is a marker of neutrophil activation, but possibly expand on the general functional properties of CD66b expressing neutrophils. There will likely be many arbovirologists reading this review and may not be completely aware of general functions of activated neutrophils. Specifically, what is the potential significance of CD66b positive neutrophils in the context of these infections.

Response:

The reviewer has raised a good point. We have modified this section as follows:

“Examining the phenotypic and functional responses of neutrophils in adult dengue patients, Opasawatchai et al. observed an upregulation of CD66b on neutrophils and early stages of NET formation, indicating an activated state, during acute DENV infection [80]. CD66b is a granulocyte activation marker involved in degranulation and production of reactive oxygen species (ROS), which is essential for antiviral activity [81,82]. Interestingly, higher levels of NET components, IL-8, and TNF-α were found in patients diagnosed with the more severe dengue hemorrhagic fever compared to patients with dengue fever or healthy controls [80].”

  1. Although Old World alphavirus infections can be associated with arthritis, the paragraph describing CHIKV infection (page 6) should include some details as to where in the body inflammatory infiltrates are seen. Again, some readers may not be completely familiar with the diseases associated with alphavirus infection. To step that out with more detail again might improve the review.

Response:

We agree with the reviewer and we have expanded this section as follows:

“CHIKV infection is primarily characterized by macrophage and monocyte infiltration into the primary sites of virus replication, which are typically the skin, muscle, and joints. However, an influx of neutrophils, dendritic cells, natural killer cells, and lymphocytes has also been observed [122].”

  1. Same paragraph on CHIKV, references 111 and 112 do not appear to be related to CHIKV.

Response:

We agree that ref 111 and 112 (ref 123 and 124 in the revised Ms) reference other viruses and we apologize for not clarifying that. This section now reads:

“Indeed, resident cells at the site of infection produce neutrophil-attracting chemokines, CXCL1 and CXCL2, following other viral infections [123,124]. This chemokine production by resident cells remains to be seen during CHIKV infection. The recruited neutrophils produce reactive oxygen species (ROS) and other cytotoxic mediators to decrease viral replication [125]. In non-mammalian models of CHIKV infection such as zebrafish, the neutrophils also serve as an important source of type I interferon for eliminating the virus and alleviating disease [126]. Even in the absence of active viral replication during chronic phases of infection, CHIKV-induced arthritis may progress due to increased cytokine expression and immune cell infiltration [122,127].”

  1. Page 6, first paragraph, last sentence needs editing. Should the second “with” be “and”?

Response:

This sentence now reads:

“CHIKV has caused reoccurring epidemics in numerous countries around the Indian Ocean since 2004 with millions of confirmed cases [103] and a surprising emergence in Europe and the Pacific Region for the first time in 2007 and 2011, respectively [104-107].”

  1. The figures are very appealing, but possibly increase font size or bold the text.

Response:

Thank you for the comment. We have increased the font size and bold the text in the figures.

Round 2

Reviewer 2 Report

Please add your comments about Ly6G with some details and explanations to the paper  to highlight the relevance of the depletion method for the readership:

"All studies referred to in this review that deplete neutrophils use Ly6G antibody treatments. While Ly6G may be transiently expressed on monocytes in the bone marrow, Ly6G antibody treatments are thought to be neutrophil-specific (Ly6 family proteins in neutrophil biology - Lee - 2013 - Journal of Leukocyte Biology - Wiley Online Library). Furthermore, ref 132 (ref 144 in the revised Ms) showed that Ly6G-mediated depletion did not affect the Ly6Chigh monocyte population but depleted the Ly6Cintermediate neutrophil population."

Author Response

Round 2 – Reviewer #2

Please add your comments about Ly6G with some details and explanations to the paper to highlight the relevance of the depletion method for the readership:

"All studies referred to in this review that deplete neutrophils use Ly6G antibody treatments. While Ly6G may be transiently expressed on monocytes in the bone marrow, Ly6G antibody treatments are thought to be neutrophil-specific (Ly6 family proteins in neutrophil biology - Lee - 2013 - Journal of Leukocyte Biology - Wiley Online Library). Furthermore, ref 132 (ref 144 in the revised Ms) showed that Ly6G-mediated depletion did not affect the Ly6Chigh monocyte population but depleted the Ly6Cintermediate neutrophil population."

Response:

Thank you for the comment. We apologize for not clarifying this in the manuscript. The end of page 8 in the revised manuscript now reads:

“It is important to note that most studies deplete neutrophils in vivo to understand their function. All the studies involving neutrophil depletion referenced in this review use Ly6G antibody treatments in mice. While Ly6G may be transiently expressed on many myeloid cells in the bone marrow including monocytes and other granulocytes, neutrophils that are circulating and recruited to the site of inflammation typically have higher Ly6G expression [148]. Basophils and eosinophils are thought to be Ly6G- or Ly6Glow/ intermediate [148]. Additionally, some studies showed that Ly6G-mediated neutrophil depletion reduced only the Ly6Cintermediate neutrophil population and not the Ly6Chigh monocyte population [144].”